# Enhancing Sustainable Waste Management Using Biochar: Mitigating the Inhibitory of Food Waste Compost from Methane Fermentation Residue on Komatsuna (*Brassica rapa*) Yield

**Nur Santi** [1], **Ratih Kemala Dewi** [2,3] , **Shoji Watanabe** [3], **Yutaka Suganuma** [4], **Tsutomu Iikubo** [4] **and Masakazu Komatsuzaki** [1,3,*]

1   United Graduate of Agricultural Science, Tokyo University of Agriculture and Technology, Tokyo 183-8509, Japan; santinur64@yahoo.co.id
2   College of Vocational Studies, IPB University, Bogor 16128, Indonesia; ratihkemala@apps.ipb.ac.id
3   College of Agriculture, Ibaraki University, Ami, Ibaraki 300-0331, Japan; watanabe@kkleaf.com
4   Hitachi Cement Co., Ltd., Hitachi 317-0052, Japan; suganuma.y@hitachicement.co.jp (Y.S.); iikubo2106@hitachi-cement.co.jp (T.I.)
*   Correspondence: masakazu.komatsuzaki.fsc@vc.ibaraki.ac.jp

**Abstract:** Methane fermentation, utilizing food waste (FW), is viewed as a sustainable strategy that leverages garbage and agricultural waste to conserve the environment. However, FW compost encounters growth inhibition issues, which we examine in this study. The aim of this study was to assess the influence of various compost mixtures on seed germination growth and the yield of Komatsuna (*Brassica rapa*). The experiment employed FW compost mixtures with biochar (BC), clay (CL), weeds (WD), and a control group in a completely randomized design with three replications to monitor germination. The experimental pots, arranged in a complete factorial design, involved three treatment factors: compost type (FW or HM), biochar presence or absence (WB or NB), and three input rates (25 g pot$^{-1}$, 50 g pot$^{-1}$, and 100 g pot$^{-1}$), each in triplicate. The combination of FW and BC exhibited an enhanced germination rate compared to FW alone. Moreover, the inclusion of biochar significantly amplified this effect, particularly at the input rate of 50 g pot$^{-1}$ and had a substantial impact on the interaction between input rate, compost type, and biochar on variables such as nitrogen (N) uptake, % N, soil carbon, and yield. Homemade BC demonstrates an increasing fertilizer cost performance (FCP) as the input rate rises across all fertilizer combinations, while commercially priced BC exhibits a reverse relationship with FCP. These findings suggest that the addition of biochar enhances the performance of methane fermentation residue compost, thereby promoting plant growth through the processing of environmentally sustainable waste.

**Keywords:** compost; food waste; biochar; komatsuna; yield

## 1. Introduction

Sustainable food systems play a pivotal role in achieving the sustainable development goals (SDGs), particularly SDG 2, which aims to eradicate hunger, ensure food security, improve nutrition, and promote sustainable agriculture. The demand for food is escalating due to the growing global population. The current food system faces significant challenges, including food waste and nutrient depletion from agriculture, leading to deteriorating water quality and soil health. These challenges are closely related to the increase in organic waste. The global generation of food waste is rising each year, causing substantial adverse effects on health, the environment, and the economy [1]. In 2018, a medium-sized anaerobic bioreactor was established to process 200 Mg of waste per day, and there are plans to develop several processes to divert food waste from landfills in the coming years [2]. To overcome these issues, the encouragement of sustainable food production through a combination of the use of organic resources is an important part.

Recycling options include the valorization of food scraps through biological conversion into high-value products such as fertilizer and bio-energy carriers like biogas and biomethane [3]. According to [4], methane fermentation compost is often challenging to use due to inhibitors and significant odor issues. However, if improvements are made to these composts, methane fermentation residue compost could attract farmer's interest. High-quality methane fermentation residue compost has the potential to contribute to soil health improvement and enhance the production of nutrient-rich, environmentally friendly plants. This can facilitate the development of a farm-to-table recycling system [4]. In accordance with the report, the primary objectives include enhancing agricultural practices, promoting clean energy, and mitigating carbon emissions [5]. Internationally, the Paris Agreement determined climate goals have initiated sustainability transitions on the agendas of numerous local, national, and global governing bodies [6]. Concurrently, the cost of renewable energy has experienced a rapid decline, making it increasingly politically and economically viable. Additionally, there has been a notable increase in public support for the urgent need to address climate change. This constitutes a critical aspect of research that aligns with the prioritization of environmental issues as per the research objectives.

Furthermore, biochar plays a crucial role in the long-term sequestration of carbon in soil, significantly contributing to climate change mitigation. In the biochar process, an initial 50% of the carbon is separated and sequestered as biochar, while the other half is converted to recoverable energy before being returned to the atmosphere. Applying larger amounts of biochar to the soil may enhance the carbon storage benefit to farmers [7].

Biochar can help establish a sustainable cycle for organic waste utilization and soil health promotion, ultimately supporting a resilient and productive food production system. Known for its high cation and anion exchange capacity, biochar enhances nutrient use efficiency when applied to soil [8]. Implementing biochar can increase soil pH, cation exchange capacity, and organic carbon levels by 46%, 20%, and 27%, respectively, with more pronounced effects observed in harsh and fine-grained soils. It is important to note that the effects on soil chemical properties may vary depending on the source material used to create the biochar [9]. However, the impact of biochar on soil conditions can be influenced by various factors, including the type and quality of the biochar, soil characteristics, crop type, rate and frequency of biochar application, and environmental conditions. Therefore, further experimentation is needed to gain a comprehensive understanding of the long-term effects of biochar on different soils and crops [10].

The combined use of biochar with fertilizers has been employed to enhance soil fertility and increase crop yields [11]. The integration of organic resources, such as animal manure and biochar, can significantly bolster sustainable food production. These practices not only improve crop yields and soil health but also contribute to the preservation of water quality and carbon sequestration, thereby making substantial progress toward the SDGs, specifically SDG 2Zero Starve. This research represents a significant contribution to the ongoing advancement of sustainable agricultural practices in Ibaraki Prefecture. It implements a system designed to recycle organic waste from the surrounding area into environmentally safe organic compost. This initiative is in alignment with the Japanese government's policy to reduce reliance on chemical fertilizers in plant cultivation and encourage the adoption of environmentally friendly organic alternatives. As per Article 16, Paragraph 1 of the Green Food System Law (Law on Promotion of Environmental Impact Reduction Business Activities to Build a Food System in Harmony with the Environment), Ibaraki Prefecture and its 44 municipalities jointly formulated a "Basic Plan for Promoting Prefectural Environmental Burden Reducing Business Activities" issued in March 2023 [12]. This plan aims to halve the use of chemical pesticides and reduce the use of chemical fertilizers by 30% by 2050. To achieve this, the ministry intends to increase the share of farmland used for organic farming in Japan from 0.5% in fiscal 2018 to 25% by 2050, expanding the area to 1 million hectares for this purpose [13].

Adopting sustainable food systems is crucial for ensuring a prosperous future for both humanity and the planet. Consequently, this study seeks to improve the attributes

of food waste (FW), explore the potential of combining FW compost with biochar, clay, and weeds in germination tests, and identify the optimal blending of FW compost with manure, such as animal manure, to enhance compost quality for Komatsuna production. The *Brasica rapa* L. species holds significant economic value worldwide, with a wide geographical distribution suitable for cultivation in both sub-tropical and tropical countries. Furthermore, the research assesses the cost-effectiveness of various compost combinations as fertilizers. This study is particularly pertinent to diverse regions grappling with waste processing challenges. However, it is important to recognize that the outcomes, especially the nutritional content of the compost produced, may vary significantly due to differences in organic waste processing methods. Hence, the regional disparities in organic waste processing represent an inherent limitation of each study.

## 2. Materials and Methods

### 2.1. Organic Waste Abundance and Process

This research utilized methane-fermented compost sourced from Tsuchiura City, Ibaraki Prefecture, Japan. FW, mainly consisting of food garbage waste, was collected daily by the Hitachi Cement Industry from various locations in the city. It was then processed at a methane gas fermentation or bio-energy factory in Ibaraki Prefecture, Japan [4]. While waste recycling systems are widespread across most regions in Japan, not all areas have facilities dedicated to processing methane fermentation bioplants. These bioplants function by recycling food-related waste, including manufacturing residues and discarded product waste from food factories, which are converted into biogas and compost through methane fermentation and fermentation composting. The biogas produced by these bioplants serves as additional fuel for eco-plants and is integrated with other facilities, particularly power plants utilizing the Feed-in Tariff (FIT) scheme for renewable energy. The electricity generated is then supplied to the community as renewable energy through electric power companies, contributing to the establishment of a recycling-oriented society in the region.

Food waste consists of household waste, discarded items from restaurants, and excess products from nearby markets close to waste processing facilities. Before being utilized in a biogas plant, the waste undergoes an initial filtration process to remove extraneous materials, such as plastic bottles, metal, and glass. Following this, the filtered waste is shredded into smaller particles, and the organic components are introduced into a fermenter tank containing methanogenic bacteria [4]. The potential abundance of agricultural waste in Tsuchiura City, Ibaraki, Japan, is illustrated in Table 1.

**Table 1.** Potential agricultural waste in Tsuchiura City, Ibaraki Prefecture, Japan.

| Agricultural Waste | Raw Material (Mg Year$^{-1}$) | Compost or Biochar (Mg Year$^{-1}$) |
|---|---|---|
| Food waste (FW) [1] | 500.0 | 6.0 |
| Animal manure (HM) [2] | 67,160.0 | 40,296.0 |
| Rice husk [3] | 743.7 | 223.1 |

[1] FW 100 Mg = 1.2 Mg year$^{-1}$; [2] HM = 23 kg day$^{-1}$, 8000 horses in Miho city; [3] rice area = 877 ha in Tsuchiura City.

The industrial process yields electrical energy and residual waste in the form of liquid and solid components. The liquid waste can be directly used as liquid organic fertilizer, while the solid components undergo further processing into solid organic fertilizer.

The quality of a fertilizer is often determined by its nutritional content, particularly the NPK (nitrogen, phosphorus, and potassium) levels. For instance, methane-fermented compost may have nutrient content values like 5.6, 1.9, and 0.5, as reported [4]. According to [14], both commercial compost and research compost exhibit sufficient nutrient content for agricultural use. In commercial fertilizer, nitrogen content ranges from 98 ppm to 2268 ppm, phosphorus from 0.871 ppm to 11,615 ppm, and potassium from 91.85 ppm to 645.55 ppm. Additionally, [15] reports dry content values for nutrient biowaste as

34.3 g kg$^{-1}$, 4.6 g kg$^{-1}$, and 5.5 g kg$^{-1}$. Typically, food waste digestate (FWD) in Europe contains nitrogen levels ranging from 1.1 kg Mg$^{-1}$ to 9.6 kg Mg$^{-1}$, phosphorus from 0.1 kg Mg$^{-1}$ to 2.4 kg Mg$^{-1}$, and potassium from 0.4 kg Mg$^{-1}$ to 2.3 kg Mg$^{-1}$. These concentrations vary significantly based on the type of food waste and the treatment process during anaerobic digestion (AD), as indicated by studies [16,17].

However, this compost encounters challenges including imbalanced nutrient content, unpleasant odor, and growth inhibition, factors that have dissuaded farmers from its adoption. In order to mitigate these external influences, the imbalanced nutritional composition of the compost impedes plant growth from germination to harvest upon application. Furthermore, the strong odor associated with the compost dampens farmers' enthusiasm for its utilization. Mixing this compost with other organic fertilizers, specifically horse manure and biochar, can enhance its quality, as depicted in Figure 1. In the picture, we can observe that this research leverages existing environmental challenges by utilizing waste collected from the surrounding area to mitigate the impact of environmental pollution. The study incorporates animal waste, addressing its contribution to water pollution, and utilizes rice product areas as materials for recycling waste into compost.

The experiment was conducted at the International Field Agricultural Center, Faculty of Agriculture, Ibaraki University, and comprised two main components. Initially, we evaluated the impact of combining FW with biochar (BC), clay (CL), and weeds (WD) on the germination of Komatsuna in May 2022. The second part focused on refining the FW and biochar combination for Komatsuna production, conducted via a pot test in June 2022.

To minimize external influences, the compost germination test was carried out in a laboratory setting, employing a completely randomized experimental design with three replications. The test involved combining FW with BC, CL, and WD, each at a ratio of 10 g of compost to 5 g of each component.

A greenhouse pot experiment was conducted over a one-month period, structured under controlled conditions and designed according to a complete factorial design with three treatment factors, each with three replications. The first factor was the type of compost, offering options of FW and horse manure (HM), serving as the primary plot. The second factor was the presence or absence of biochar, represented by options "with biochar" (WB) and "without biochar" (NB). The final factor was the input rate of compost, with three different rates per pot (25 g pot$^{-1}$, 50 g pot$^{-1}$, and 100 g pot$^{-1}$), conducted in triplicate. The compost combinations tested in the experiment included: FW + HM (1:1), FW + BC (1:1), HM + BC (1:1), FW + HM + BC (1:1:1), and a control group (CT) with no compost addition.

*2.2. Assessment of Compost Quality*

2.2.1. Germination Test

A germination test of Komatsuna was performed using hot water extracts derived from a mix of compost components, including FW, BC, CL, and WD. Each compost component weighed 10 g, and 5 g each of BC, CL, and WD were added to a 100-mL erlenmeyer flask. Following this, 100 mL of hot water was added, and the mixture was stirred for 30 min. The mixture was then filtered using a suction pump vacuum, and the resulting extract was transferred to a 50-mL Corning centrifuge tube.

Komatsuna seeds were sown in Petri dishes, with each dish housing 50 seeds. Each Petri dish was treated with 2 mL of the extract liquid fertilizer. The seeds were then incubated at a temperature of 24 °C. After a germination period of two days, the germination rate was recorded, and the length of the radicle was measured. The evaluation of radicle length is sensitive to the inhibitory effects of the compost.

2.2.2. Ion Evaluation of the Compost

Quality evaluation was conducted using a Metrohm Ion Chromatography (Eco IC) and an 863 Compact Autosampler capable of analyzing the presence of anions and cations in each sample. Ion chromatography, also known as ion-exchange chromatography, is a chromatographic technique used to separate ions and polar molecules based on their

affinity to the ion exchanger. This separation relies on the electrical charge of the protein, which is determined by both positively and negatively charged chemical groups that can be altered by adjusting the pH of the buffer solution. The process employs a gradient with a linear increase in salt concentration [18]. Eco ICs are specifically designed for routine analysis of anions, cations, and organic acids. They incorporate essential components such as chemical suppressors, conductivity detectors, and dedicated software, enabling comprehensive analytical capabilities. Alternatively, another technique involves using an ion chromatograph equipped with a PU-2080i plus high performance licuid chromatography (HPLC) pump. However, this represents an older version compared to the Eco IC [19].

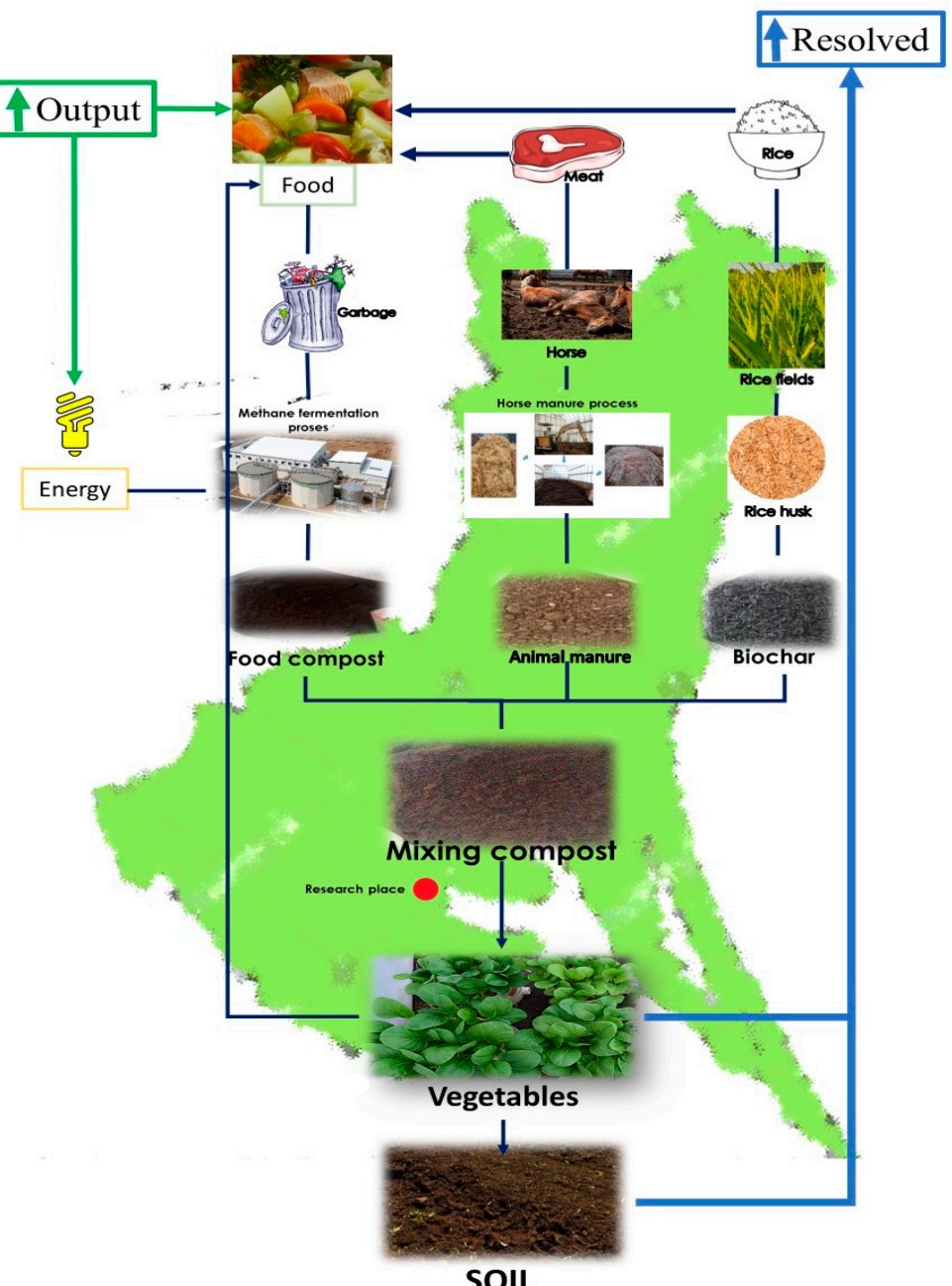

**Figure 1.** Agricultural waste and its utilization in Tsuchiura City, Ibaraki, Japan.

The sample for analysis was prepared by extracting compost to create a sample solution. Specifically, 0.5 mL of the sample was diluted with 50 mL of distilled water. The resulting diluted solution was then transferred into a 10-mL container using a 3-mL

filter syringe 0.22 μm. The selected compost sample for further analysis was introduced into the ion chromatography machine through the analysis port. For the analysis of anion content, a solution composed of sodium carbonate and sodium bicarbonate was used as the eluent. Additionally, a 100 mM $H_2SO_4$ solution was used as part of the process to identify anion content. For the analysis of cation content, nitric acid and dipicolinic acid were utilized. Standard anion and cation contents served as reference ions during the analysis of each sample.

*2.3. Greenhouse Experiment*

2.3.1. Pot Experiment Design

The Komatsuna pot experiment was carried out in a greenhouse using various compost media. These included combinations of FW with HM with a ratio of 1:1 (FW + HM), FW with biochar (BC) in a 1:1 ratio (FW + BC), HM with BC in a 1:1 ratio (HM + BC), FW with HM and BC in a 1:1:1 ratio (FW + HM + BC), and a control group (CT) with no compost addition. These combinations were tested at different compost input rates (25 g pot$^{-1}$, 50 g pot$^{-1}$, and 100 g pot$^{-1}$).

The cultivation test, which lasted one month, was conducted in Wagner pots measuring 17 cm in diameter and 20 cm in height. The pots were filled with *kousitu kanumatuti* soil, which contained medium and large granules. Prior to seed planting, the plant media were prepared according to the treatment specifications and then added to the pot. Each pot was equipped with five holes, with three seeds sown in each hole [4].

After one month, the Komatsuna was ready for harvest. The harvest time was determined by cutting the vegetables (excluding the roots), and the fresh weight of the plant was measured. The harvested Komatsuna was then dried for three days at 60 °C in an oven. The dry weight of the plant was subsequently measured. The yield of Komatsuna was calculated using the fresh weight of Komatsuna per pot, and the dry weight of Komatsuna was used to calculate nitrogen (N) uptake.

2.3.2. Komatsuna Biomass and Planting Medium Measurements

The carbon (C) and nitrogen (N) content of both the Komatsuna and the planting medium were determined using a CN analyzer (JM 3000, J-Science Lab, Kyoto, Japan) through dry combustion methods. For the analysis, approximately 100–200 mg of finely ground Komatsuna was placed on the sample mount. After the complete burning process, the C and N contents were obtained. Following this, the nitrogen uptake and nitrogen use efficiency (NUE) were calculated. N uptake was calculated using the following formula:

$$N\ uptake = N\ (\%) \times dry\ biomass\ content\ of\ Komatsuna \tag{1}$$

Then, NUE was determined as the ratio of Komatsuna yield to the applied nitrogen using the following formula:

$$NUE = \frac{(Yield\ a\ -\ Yield\ 0)}{(Input\ N)} \tag{2}$$

$$Input\ N = Input\ level \times N(\%) \tag{3}$$

In this context, "yield a" represents the yield of Komatsuna under a specific compost treatment, while "yield 0" denotes the yield of Komatsuna in the control treatment. "Input N" refers to the quantity of nitrogen applied to the plants [4]. The calculated nitrogen input for each treatment depends on the compost input levels used in this study: 25 g pot$^{-1}$, 50 g pot$^{-1}$, and 100 g pot$^{-1}$. The cost of C was calculated using the subsequent equation:

$$C = \frac{(\%\ C\ fertilizer\ -\ \%\ C\ ct)}{(Cost\ in\ USD)} \tag{4}$$

In this equation, "% C fertilizer" refers to the percentage of carbon content in the soil where Komatsuna is grown, "% C ct" denotes the carbon content of the control sample, and "Cost in USD" represents the cost to produce carbon.

### 2.4. Fertilizer Cost Perform (FCP) Analysis of Combination FW with Biochar Effect

Fertilizer cost performance (FCP) is a measure used to evaluate the cost-effectiveness of fertilizers. It is calculated by dividing the increase in yield by the cost of the fertilizer. A higher FCP value signifies a more cost-effective fertilizer. FCP is a useful tool for farmers, aiding them in selecting the most appropriate fertilizers for their specific crops and soil conditions. The formula for calculating FCP is as follows:

$$FCP = \frac{Yield\ (fertilizer) - Yield\ (CT)}{Cost\ (fertilizer)} \tag{5}$$

Yield (fertilizer) = yield in certain compost g pot$^{-1}$, yield CT = yield in control treatment g pot$^{-1}$, cost (fertilizer) = cost of fertilizer USD g$^{-1}$.

### 2.5. The Economic Analysis b/c Ratio

The economic feasibility was assessed using a benefit-to-cost ratio (B/C ratio). B/C ratio exceeding 1 indicates that the management is economically viable. The total cost and benefit components were calculated using the following formula:

$$Total\ cost\ (USD) = Variable\ cost + Depreciation\ cost \tag{6}$$

$$Income\ (USD) = Yield\ (komatsuna) \times Price\ (komatsuna) \tag{7}$$

$$Profit\ (USD) = Income - Total\ cost \tag{8}$$

$$B/C\ Ratio = \frac{Total\ cost}{Profit} \tag{9}$$

In this equation, variable costs consist of materials, fuel, transportation, electricity, and labor. Depreciation costs encompass the investment cost per year, which can be found in homemade biochar. In the analysis of economic implications associated with the integration of biochar, a simulation of agricultural conditions is utilized. This comprehensive analysis considers both commercial and homemade biochar, taking into account various factors such as costs, revenues, and cash flows involved in farming practices. Homemade biochar has no price as it is produced by farmers themselves, while commercial biochar must be purchased. The calculation of cost factors involves aggregating costs per planting season per hectare unit. This calculation includes material costs, daily labor expenses at an hourly rate, and transportation expenses for materials per planting season.

### 2.6. Statistical Analysis

Analysis of variance was performed to analyze the mixing of FW compost, HM compost, and BC in germination and pot experiment at $p < 0.05$ significance level followed by the Tukey–Kramer post hoc test using the ©Statview 5.0.1 software.

## 3. Results

### 3.1. Komatsuna Seed Germination Is Affected by Different Mixing Compost

#### 3.1.1. Vegetation of Komatsuna

Figures 2 and 3 presents the results for Komatsuna seeds, showcasing the total radicle length, germination percentage, GI and ammonia content when combined with various components, including biochar, clay, and weed. Among these combinations, the highest germination rates of 100% and 76.6% were observed in the CT and the biochar FW with BC combination, respectively. Conversely, the lowest germination percentage was recorded for FW + WD, with a mere 52% germination level.

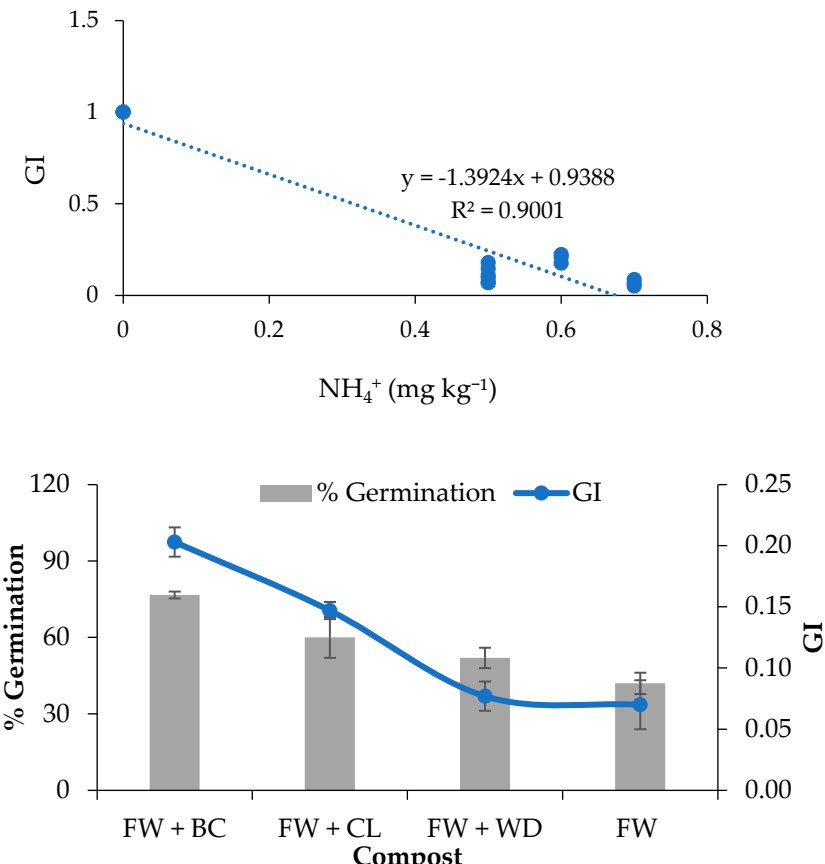

**Figure 2.** Variation in the germination index (GI) in relation to the $NH_4^+$ content for each compost treatment. It also includes the germination percentage of Komatsuna seeds. The compost treatments are as follows: FW represents food waste, BC stands for biochar, CL denotes clay, and WD signifies weed.

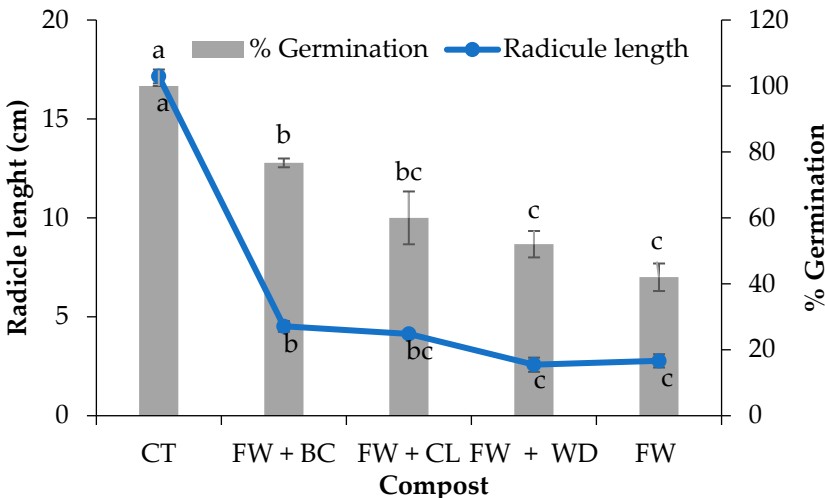

**Figure 3.** The germination percentage and radicle length of Komatsuna seeds affected by different compost treatments. FW: food waste; BC: biochar; CL: clay; WD: weed; CT: control. Different lowercase letters indicate significant differences between compost treatments based on Tukey–Kramer test at $p < 0.05$.

The radicle length of the germinated seeds was also influenced by the combination of compost and biochar. As depicted in Figure 3, the combination that resulted in the

longest radicle length was FW + BC, measuring 4.5 cm. Interestingly, this length was not significantly different from the radicle length of 4.1 cm observed in the FW + CL combination. In contrast, the shortest radicle length of 2.5 cm was found in the FW + WD combination, while the food waste compost alone exhibited a radicle length of radicle cm.

The highest GI for Komatsuna was achieved with the FW + BC combination, registering an index value of 0.20. Interestingly, this result was not significantly different from the GI of 0.14 observed in the FW + CL combination. On the other hand, the lowest GI values were associated with the FW + WD combination and the single compost FW combinations, with indexes of 0.08 and 0.07, respectively.

This study emphasizes that the use of FW compost alone yielded lower results compared to the combination of FW and BC compost, which was effective in improving the germination rate and radicle length of seeds. Therefore, these findings suggest that incorporating BC into FW compost could significantly enhance both the germination percentage and radicle length of seeds.

### 3.1.2. Properties of FW Compost

Table 2 illustrates the presence of anions and cations in various compost combinations, particularly those involving FW with BC, CL, and WD. In the FW compost, the $Cl^-$ content was 0.5 g $kg^{-1}$, which was lower than that in the FW + BC, FW + CL, and FW + WD combinations, where the $Cl^-$ levels were 0.6 and 0.7 g $kg^{-1}$, respectively. The $NO_3^-$ content in all combinations was lower than all other contents. Additionally, the $SO_4^{2-}$ content in the FW + CL combination was notably higher at 0.6 g $kg^{-1}$ compared to that in single FW, FW + WD, and FW + BC, which had respective values of 0.5 g $kg^{-1}$.

**Table 2.** Anion and cation content in the compost.

| Compost | Anion (g $kg^{-1}$) | | | Cation (g $kg^{-1}$) | | | | |
|---|---|---|---|---|---|---|---|---|
| | $Cl^-$ | $NO_3^-$ | $SO_4^{2-}$ | $Na^+$ | $NH_4^+$ | $K^+$ | $Ca^{2+}$ | $Mg^{2+}$ |
| FW | 0.5 | 0.0 | 0.5 | 0.3 | 0.7 | 0.2 | 0.4 | 0.1 |
| FW + WD | 0.7 | 0.0 | 0.5 | 0.3 | 0.5 | 0.7 | 0.3 | 0.1 |
| FW + CL | 0.6 | 0.0 | 0.6 | 0.6 | 0.5 | 0.2 | 0.4 | 0.1 |
| FW + BC | 0.6 | 0.0 | 0.5 | 0.3 | 0.6 | 0.4 | 0.4 | 0.1 |

Note: FW (food waste); BC (biochar); CL (clay); and WD (weeds).

On the contrary, the $Na^+$ content showed a significant increase in the FW + CL combination, reaching 0.6 g $kg^{-1}$. In a single FW scenario, the $NH_4^+$ content, measuring 0.7 g $kg^{-1}$, exceeded that observed in all compost combinations. The $K^+$ content in the FW + WD combination peaked at 0.7 g $kg^{-1}$, surpassing other combinations. In contrast, the $Ca^{2+}$ content consistently remained lower across compost combinations, measuring 0.3 g $kg^{-1}$ compared to all other combinations. Conversely, the $Mg^{2+}$ content in all combinations remained constant, recording 0.1 g $kg^{-1}$, respectively.

### 3.1.3. Relationship between Ammonia, Sulfate Content, and Percent Germination with GI

Table 3 presents data on the GI in relation to the ammonia and sulfate content, as well as the germination percentage in compost.

A negative correlation is evident between GI and ammonia content, suggesting that higher levels of ammonia in the compost negatively impact GI. Notably, both FW compost and the FW with WD combination display significantly lower GIs compared to other conditions. This finding strongly suggests that the elevated ammonia content within the compost adversely affects the germination process of Komatsuna plants.

In contrast, varying GIs are observed in the cases of FW with BC and FW with CL. The positive influence of sulfate content on Komatsuna germination becomes apparent, explaining the higher GI in the FW + BC and FW + CL combinations compared to the FW + WD combination and the single FW condition. The single FW condition demonstrates

notably lower germination percentages and GI, whereas the FW combinations show the opposite trend (as depicted in Figure 2). Specifically, the combination of FW with BC exhibits significantly higher germination percentages and GIs than the single FW condition, a pattern consistent across the other combinations.

**Table 3.** Nitrogen percentage (%) of media and Komatsuna at different input rates and compost.

| Compost | Input Rate | Biochar | % N of Komatsuna | % N of Media |
|---|---|---|---|---|
| CT | | NB | 1.51 ± 0.30 fgh | 0.00 ± 0.00 k |
| | | WB | 1.16 ± 0.09 gh | 0.00 ± 0.00 k |
| FW | 25 | NB | 2.69 ± 0.30 d | 1.76 ± 1.76 cd |
| | | WB | 1.70 ± 0.14 defgh | 0.55 ± 0.55 ghijk |
| | 50 | NB | 3.99 ± 0.04 c | 2.38 ± 2.38 bc |
| | | WB | 2.64 ± 0.09 de | 1.08 ± 1.08 efgh |
| | 100 | NB | 7.33 ± 0.05 a | 5.33 ± 5.33 a |
| | | WB | 5.38 ± 0.12 b | 2.24 ± 2.24 bc |
| HM | 25 | NB | 1.11 ± 0.01 gh | 0.57 ± 0.57 ghijk |
| | | WB | 0.98 ± 0.02 h | 0.17 ± 0.17 jk |
| | 50 | NB | 1.62 ± 0.17 efgh | 0.83 ± 0.83 fghij |
| | | WB | 1.38 ± 0.20 fgh | 0.29 ± 0.29 ijk |
| | 100 | NB | 2.29 ± 0.09 def | 1.27 ± 1.27 def |
| | | WB | 1.65 ± 0.18 defgh | 0.48 ± 0.48 hijk |
| FW + HM | 25 | NB | 1.33 ± 0.04 fgh | 0.73 ± 0.73 fghij |
| | | WB | 1.77 ± 0.16 defgh | 0.90 ± 0.90 efghi |
| | 50 | NB | 2.09 ± 0.10 defg | 1.55 ± 1.55 de |
| | | WB | 2.34 ± 0.16 def | 1.23 ± 1.23 defg |
| | 100 | NB | 3.82 ± 0.39 c | 2.49 ± 2.49 b |
| | | WB | 5.10 ± 0.37 b | 2.67 ± 2.67 b |
| | ANOVA | | | |
| | compost | | *** | *** |
| | input rate | | *** | *** |
| | biochar | | *** | *** |
| | Compost × input rate | | *** | *** |
| | Biochar × compost | | *** | *** |
| | Biochar × input rate | | ns | *** |
| | biochar × compost × input rate | | *** | *** |

Note: CT (control); FW (food waste); BC (biochar); HM (horse manure); *** indicates significant difference at *p* < 0.001; ns: not significant; value = mean ± standard error. The presence of different lowercase letters indicates significant differences between the compost treatments, input rates, and biochar treatments different at based Tukey–Kramer test.

### 3.2. Komatsuna Production Is Affected by Mixing FW with HM and the Addition of BC

3.2.1. Yield of Komatsuna

Figure 4 depicts the variations in Komatsuna yields under different compost treatments and input levels. Compared to the control, the combination of FW and HM demonstrated an increase in yield at an input rate of 100 g pot$^{-1}$.

Furthermore, the addition of BC was found to significantly enhance yield at an input rate of 50 g pot$^{-1}$. It's noteworthy that the increased yield was achieved with the FW + HM combination at an input rate of 100 g pot$^{-1}$. The incorporation of BC positively impacted yield when combined with the FW + HM treatment at an input rate of 50 g pot$^{-1}$. However, it's crucial to note that the supplementation of BC did not result in a similar increase in yield for the other treatment conditions.

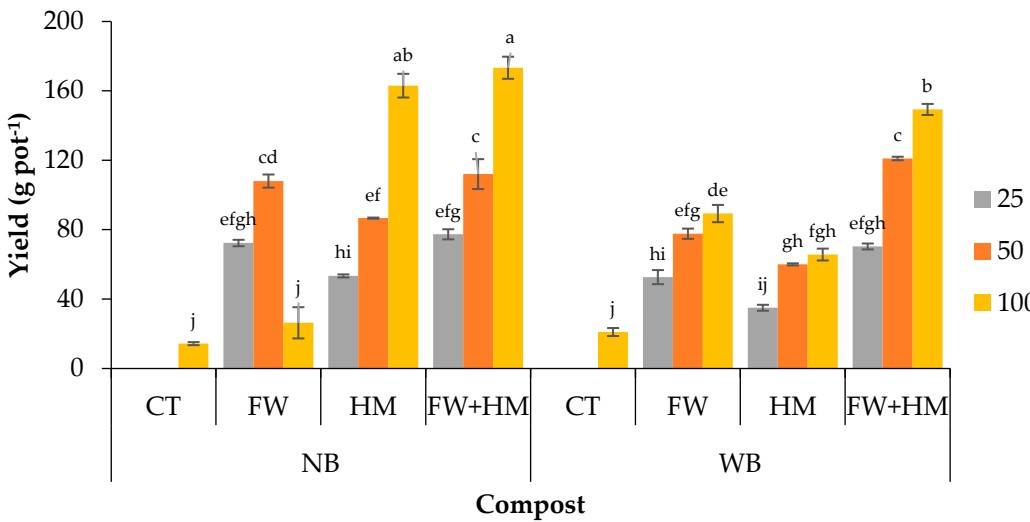

**Figure 4.** Impact of various compost treatments and input rates on the yield of Komatsuna. FW: food waste; HM: horse manure; CT: control; NB: no biochar; WB: with biochar. The presence of different lowercase letters indicates significant differences between the compost treatments, input rates, and biochar treatments based on Tukey–Kramer test at $p < 0.05$.

### 3.2.2. Carbon and Nitrogen Content of the Planting Medium Komatsuna

As depicted in Figure 5, the use of FW in compost demonstrates the potential to increase soil carbon content across various input rates. Notably, the inclusion of BC in this treatment shows enhanced effectiveness, particularly at an input rate of 100 g pot$^{-1}$. Furthermore, high levels of soil carbon were observed in the FW and HM combination at an input rate of 100 g pot$^{-1}$. The addition of BC had a positive impact on soil carbon content within the FW + HM treatment at the same input rate. However, it's important to note that the addition of BC did not result in increased soil carbon levels for all treatment conditions. Additionally, no significant interaction was observed between input rate, compost application, and BC with respect to soil carbon input levels.

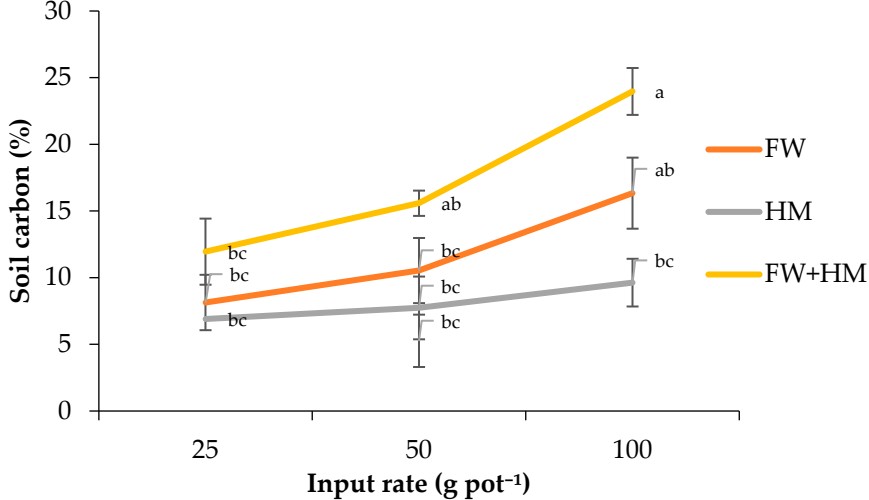

**Figure 5.** Soil carbon content in komatsuna media-affected soil carbon by different input rates and compos. FW: food waste; HM: horse manure; different lowercase letters indicate significant differences between compost, input rate, and biochar treatments based on Tukey–Kramer test at $p < 0.05$.

### 3.2.3. C Cost of Komatsuna Production

Figure 6 presents data on carbon costs, revealing that the application of FW combinations led to an increase in carbon costs across all input rates. Notably, the inclusion of BC in this treatment showed significant effectiveness, particularly at an input rate of 100 g pot$^{-1}$. The highest carbon cost was observed in the FW and horse manure (FW + HM) combination at an input rate of 100 g pot$^{-1}$, while the smallest was recorded in the HM treatment at 25 g pot$^{-1}$. The addition of biochar had a positive impact on carbon costs within the FW + HM treatment at an input rate of 100 g pot$^{-1}$. However, it's important to emphasize that BC did not increase carbon costs within the other treatment groups. Consequently, no significant interaction was observed among input rate, compost, and BC in this context.

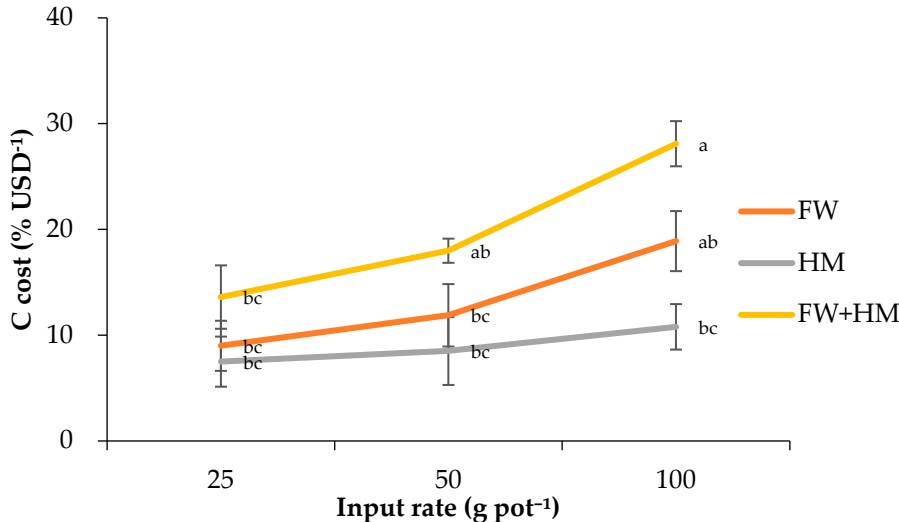

**Figure 6.** Carbon cost of komatsuna production affected by different input rates and compost. FW: food waste; HM: horse manure; different lowercase letters indicate significant differences between compost, input rate, and biochar treatments based on Tukey–Kramer test at $p < 0.05$.

### 3.2.4. Nitrogen Uptake and Nitrogen Use Efficiency (NUE)

In terms of nitrogen content (% N), compared to the control group, the application of FW led to increased % N levels in both Komatsuna and the growing medium across all input rates. Notably, the addition of BC to the FW treatment showed enhanced effectiveness, especially at an input rate of 100 g pot$^{-1}$. Moreover, the highest % N concentration was observed in the FW treatment at an input rate of 100 g pot$^{-1}$, while the smallest % N concentration was reported in the HM treatment at an input rate of 25 g pot$^{-1}$. It's worth noting that the addition of BC had a positive influence on % N levels within the FW + HM treatment at an input rate of 100 g pot$^{-1}$. Nonetheless, it's important to highlight that BC did not lead to an increase in % N within the other treatment groups, as shown in Figure 7.

Figure 8 presents data on NUE, emphasizing that the use of FW combinations enhances NUE across all input rates. Notably, the addition of biochar (BC) to this treatment shows increased effectiveness, especially at an input rate of 100 g pot$^{-1}$.

Furthermore, the highest NUE values were observed in the FW and HM combination at an input rate of 100 g pot$^{-1}$, while the lowest values were reported in the single FW treatment at the same input rate. The inclusion of BC had a positive impact on NUE within the FW + HM treatment at an input rate of 100 g pot$^{-1}$. However, it's important to note that the addition of BC did not lead to an increase in NUE within the other treatment groups.

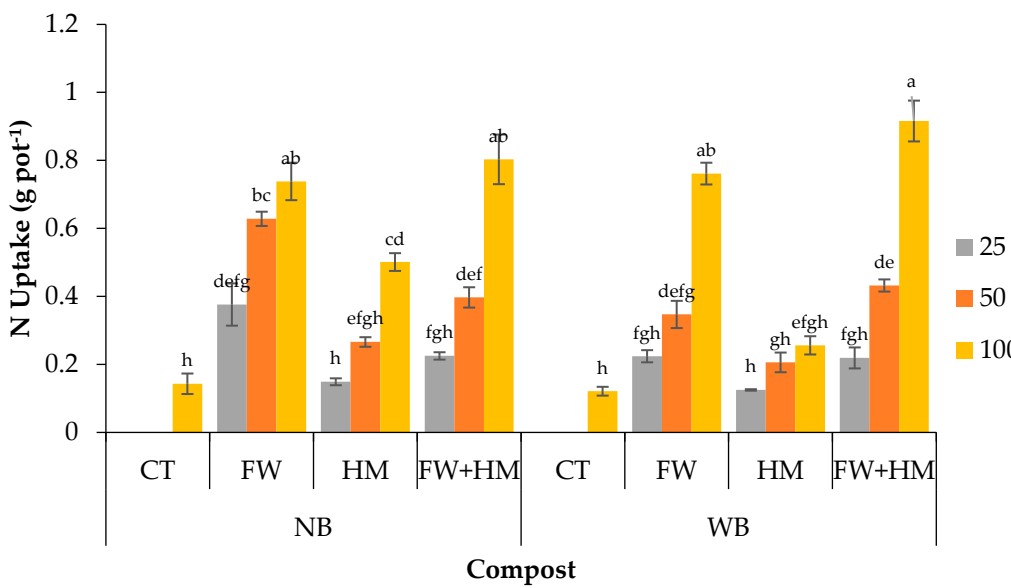

**Figure 7.** Nitrogen (N) uptake of komatsuna affected by different input rates and compost. FW: food waste; HM: horse manure; CT: control; NB: no biochar; WB: with biochar. Different lowercase letters indicate significant differences between compost, input rate, and biochar treatments based on Tukey–Kramer test at $p < 0.05$.

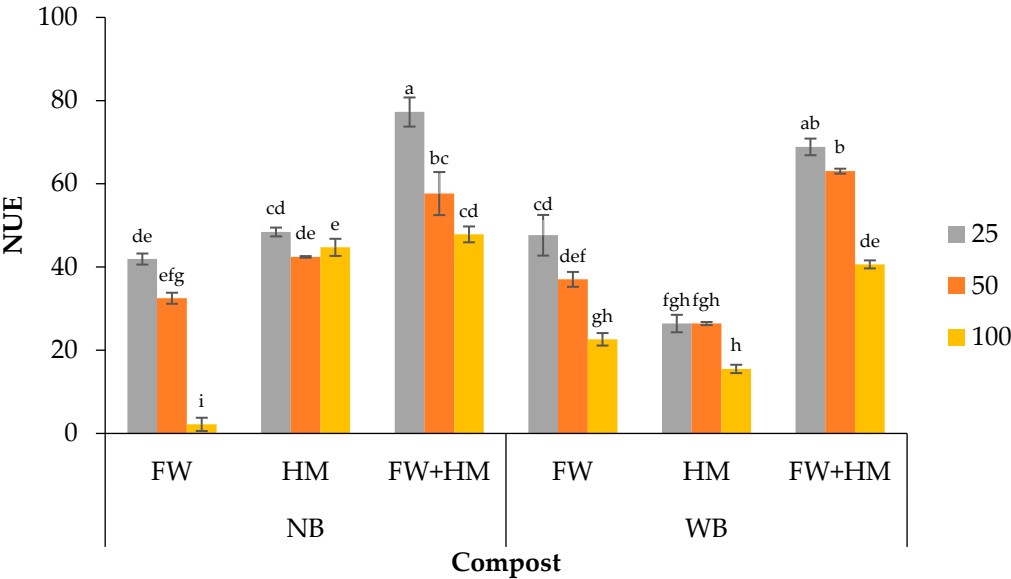

**Figure 8.** Nitrogen use efficiency (NUE) of komatsuna affected by different input rates and compost. FW: food waste; HM: horse manure; CT: control; NB: no biochar; WB: with biochar. Different lowercase letters indicate significant differences between compost, input rate, and biochar treatments based on Tukey–Kramer test at $p < 0.05$.

### 3.2.5. Economic Cost/Benefit Analysis of Compost

In Table 4, a comprehensive cost–benefit analysis of compost utilization in the context of Komatsuna production is presented. The cost associated with the compost exhibits variability within the range of 0.1 to 1.1 USD kg$^{-1}$. Specifically, the cost of utilizing compost derived from FW or HM is approximately 0.1 USD kg$^{-1}$, aligning with prevailing market prices. Notably, this cost is significantly lower than that of biochar, priced at around 0.9 USD kg$^{-1}$. However, when considering combined applications with other organic fertilizers such as FW + BC, HM + BC, and FW + HM + BC, the overall cost of fertilizer utilization increases to approximately 1.1 USD kg$^{-1}$.

**Table 4.** Compost prices based on the combination of compost.

| Fertilizer | Price (USD kg$^{-1}$) |
|---|---|
| BC | 0.98 |
| FW | 0.14 |
| HM | 0.14 |
| FW + HM | 0.14 |
| FW + BC | 1.12 |
| HM + BC | 1.12 |
| FW + HM + BC | 1.12 |

Note: FW, food waste; HM, horse manure; BC, biochar.

Figure 9 presents the FCP for various input levels involving different fertilizers, including FW, HM, BC, FW + BC, HM + BC, and FW + HM + BC. In evaluating the economic analysis related to mixing biochar with various composts, we compared harvest yields across treatments and then multiplied the unit price for each type of fertilizer. This analysis takes into account factors such as the unit price of fertilizer and the yield of each treatment relative to the control yield. However, a crucial consideration is quantifying the fertilizer input provided to the plant in relation to the achieved harvest. This evaluation enables us to determine the effectiveness of fertilizer utilization performance relative to the price of fertilizer. In Figure 9A, the FCP for BC derived from homemade sources exhibits a decreasing trend as input quantities increase across all fertilizer combinations. Evidently, BC-based combinations display optimal cost performance at lower input levels. Conversely, the HM + BC combination shows less favorable FCP when input rates are increased. Subsequently, Figure 9B illustrates that when considering commercially available BC, the FCP exhibits an increasing trend as input rates decrease across all fertilizer combinations.

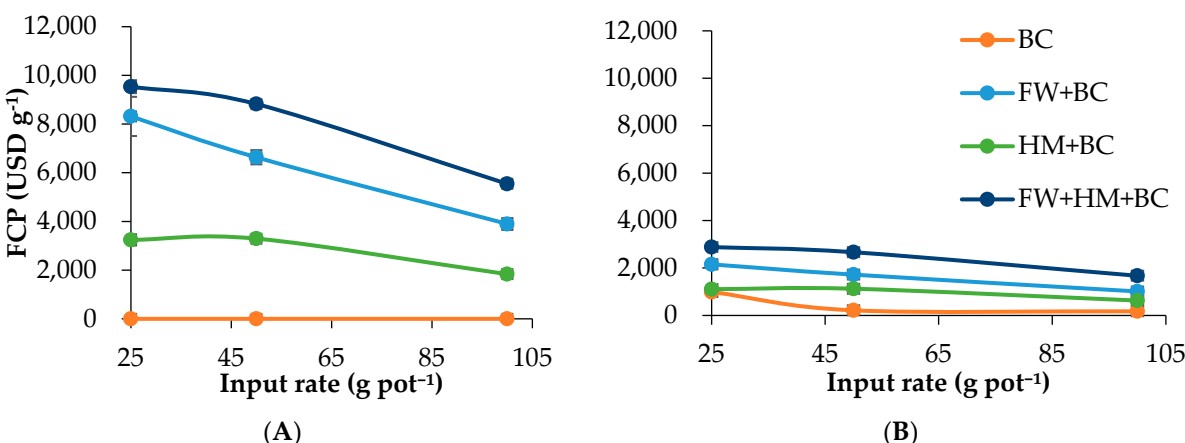

**Figure 9.** Fertilizer cost performance (FCP): (**A**) Biochar homemade price; and (**B**) Biochar commercial price; FW: food waste; HM: horse manure; BC: biochar.

Figure 10 provides a graphical representation of the economic cost analysis of the BC system, encompassing various components such as material costs, fuel expenses, labor costs, electricity consumption, transportation costs, income, profit, and the benefit-to-cost (B/C) ratio. In examining the economic implications associated with the integration of biochar, we utilize a simulation of agricultural conditions. This simulation replicates planting procedures conducted within a controlled greenhouse on a 1-hectare agricultural land area. The analysis considers both commercial and homemade biochar, taking into account various factors such as costs, revenues, and cash flows involved in farming practices. Cost factors involved in the calculation, including materials, labor, and transportation, are computed by aggregating costs per planting season per hectare unit. This encompasses

material costs necessary for one planting season, labor expenses incurred per day, hourly work costs, and transportation expenses for materials per planting season.

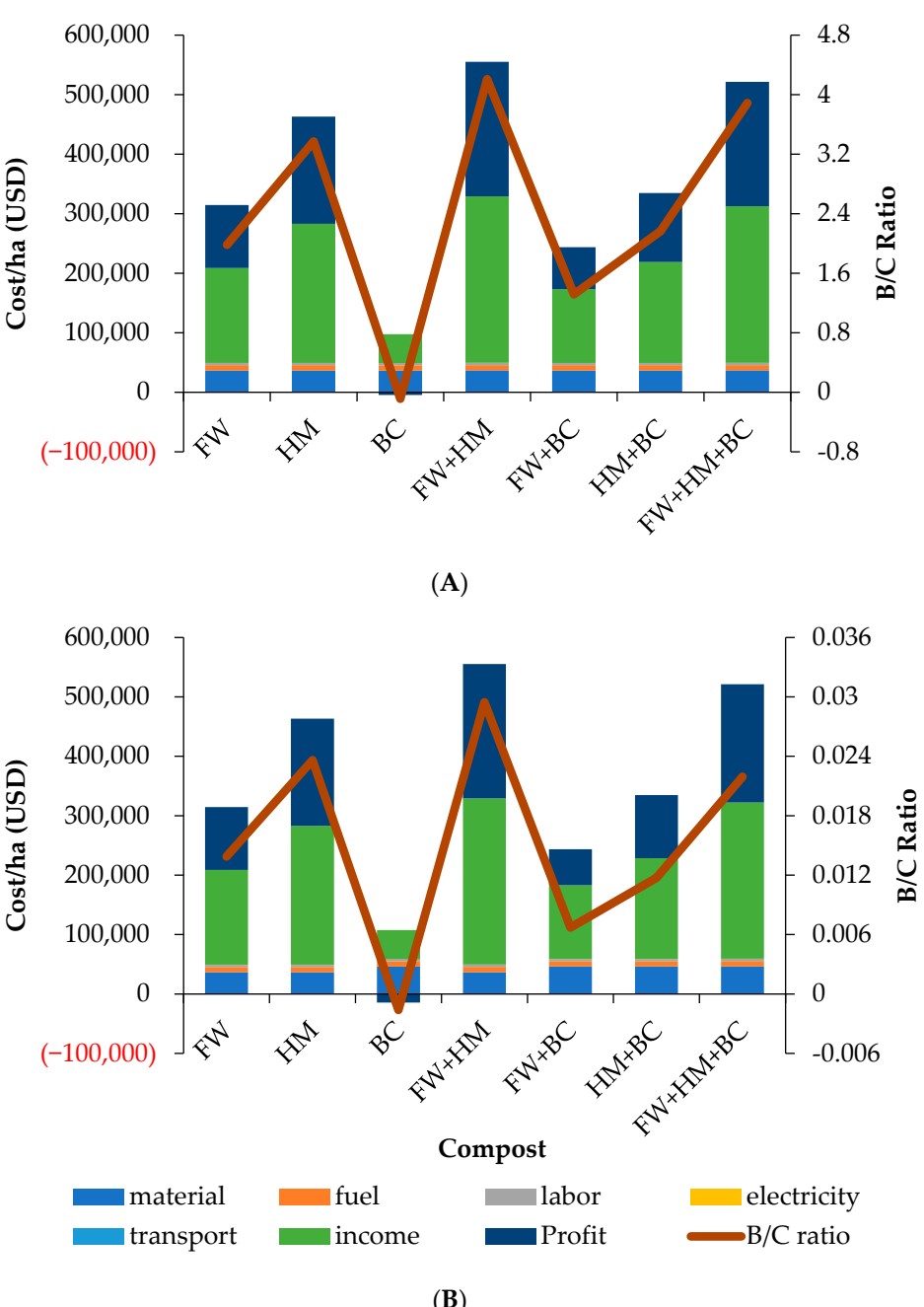

**Figure 10.** Economic cost analysis of biochar systems: (**A**) Biochar homemade price; and (**B**) Biochar commercial price; FW: food waste; HM: horse manure; BC: biochar.

As shown in Figure 10A, when homemade-produced biochar is incorporated into the application of a single FW compost, it potentially increases both the cost of profit and the B/C ratio compared to scenarios involving single FW compost. However, it outperforms single FW compost in Figure 10B. Furthermore, it's noteworthy that the use of homemade-produced BC consistently outperforms its commercially sourced counterpart in terms of profit and the B/C ratio.

## 4. Discussion

### 4.1. Komatsuna Seed Germination under Different Combinations of FW with Biochar, Clay, and Weed

Composting is a natural biological process where organic materials such as twigs, grass, flowers, and remains of fruits and vegetables are transformed into humus-rich soil. This serves as a valuable resource for replenishing nutrients and bio elements in the soil [20]. The quality of compost, including factors like reliability, should be evaluated prior to its application in the field. Unstable or immature compost can have a detrimental impact on seed germination, vegetative growth, and the soil habitat due to reduced oxygen stock, limited available nitrogen, or the presence of phytotoxic compounds [21].

A previous study by [4] reported that combining compost from methane fermentation residue with animal manure could enhance the of quality of compost. In this study, we also utilized a combination of compost derived from methane fermentation residue with biochar, clay, and weeds in germination tests and pot experiments, demonstrating its suitability for potted plant growth. One method to evaluate the quality of compost concerning germination is by examining radicle length during germination. Assessing radicle length during the germination process is vital for determining the nutritional efficacy of the provided fertilizer for seed germination. The growth of radicle length during germination provides evidence for identifying potential growth-inhibiting factors in the plant. According to [22], the extension of the radicle through the surrounding embryo structure signifies the completion of germination and the beginning of seedling growth. Consequently, the measurement of radicle length is crucial in this research to ascertain the effectiveness of the germination process.

In the present study, notable disparities in the percentage of Komatsuna seed germination were observed between the use of FW compost, with a germination rate of 40%, and the FW + BC combination, which achieved a significantly higher germination rate of 76% (Figure 3). This variance can be ascribed to the low anion content, particularly chlorine, in FW compost, with FW containing 0.5 g kg$^{-1}$ of chlorine (Table 2). Underscored the significance of chlorine as an essential micronutrient for plants, positively influencing crop yields and quality when adequately supplied. However, an excess of chlorine can contribute to salinity stress and plant toxicity [23].

Furthermore, FW compost exhibited a high ammonia content, reaching 0.7 g kg$^{-1}$ (Table 2). Figure 2 presents an analysis illustrating the relationship between germination percentage, germination index rate, and ammonia concentration. Research by [24] on orchid seeds indicated that ammonia content at a concentration of 63.7 mg/L stimulated growth and development, but higher concentrations gradually led to a decrease in germination, with total inhibition observed at 510 mg L$^{-1}$. Excessive ammonium accumulation beyond the requirements of seed metabolism can be toxic, impeding seed growth and development [25]. Ammonium toxicity results from the disruption of the cell membrane's charge gradient, allowing ammonia to penetrate the membrane and inhibit photophosphorylation [26]. Ammonium also affects root growth, gravitropism, and auxin transporters in plants [27]. At a concentration of 1 mM, ammonium has been shown to inhibit primary root growth, reducing both elemental expansion and cell production. It also hampers the length of the elongation zone and the maximum elemental expansion rate, while decreasing the apparent length of the meristem and the number of dividing cells without affecting the cell division rate. Additionally, ammonium reduces the number of root cap cells but does not seem to impact the status of the root stem cell niche or the distal auxin maximum at the quiescent center. Furthermore, it inhibits root gravitropism and concomitantly down-regulates the expression of two pivotal auxin transporters, AUX1 and PIN2 [28]. The germination outcomes of compost derived from methane fermentation indicate the presence of growth inhibitors within this compost. This observation correlates with the elevated ammonia content, identified as a growth-inhibiting factor in the compost. Consequently, alternative compost mixing methods show the potential to mitigate these growth-inhibiting factors. This observation aligns with the germination results across all treatments in the

study, where consistent improvements in germination, including radicle growth, were observed compared to compost derived solely from methane fermentation. The optimal results observed in the germination test of this study revealed that blending compost produced through methane fermentation with biochar effectively countered the growth inhibitors present in the methane fermentation-derived compost. This indicates that such a combination can balance the nutrient profile of the compost, providing farmers with a more accessible and relatively affordable alternative.

*4.2. Improvement FW of Compost for Komatsuna Production*

In the Komatsuna growth test conducted in a greenhouse using pots, it was observed that the combination of FW compost with biochar significantly increased Komatsuna yields, as evidenced by the post-harvest yield increase. This improvement in yield is also reflected in Figure 4, which demonstrates that the quality of compost derived from methane fermentation can be enhanced when applied to plants in combination with biochar. Furthermore, nutrient deficiencies in methane-fermented fertilizer can be effectively mitigated when combined with other organic materials [4]. Notably, the more compost combination fertilizers derived from waste and biochar are used, the higher the plant growth rate compared to using FW as a single fertilizer. Additionally, the addition of biochar can reduce the presence of other growth-inhibiting species (inhibitors) in the soil, thereby increasing yields when compared to unamended soil systems [29].

Figure 5 presents soil carbon data, indicating statistical significance within compost, input rate, and biochar factors. However, it's important to note that no statistically significant interaction was observed among all three factors. Instead, statistical significance was found solely in the interaction between compost and input rate concerning soil carbon levels. As reported by [30], higher biochar application rates lead to a greater increase in total carbon, ranging from 28.9% for rates $\leq$ 1% to 140% for rates > 5%. Therefore, soils with less than 1% total C before biochar implementation showed a higher percentage enhancement than soils with high initial total C (>2%).

Regarding carbon costs, as shown in Figure 6, a notable interaction between input levels and compost is evident in relation to carbon costs across all treatments. Conversely, the introduction of biochar has a significant influence on carbon costs, albeit in specific aspects of the treatment. Additionally, there is an absence of statistically significant interaction among the three variables: input rate, compost, and biochar. Remarkably, carbon costs exhibit a substantial increase, reaching up to 28 USD per unit in the combination of FW and HM at an input rate of 100 g pot$^{-1}$.

When it comes to carbon cost analysis, the results mirror the levels of carbon in the soil. Notably, the FW + HM combination at all input rates shows a significant increase in soil carbon levels, leading to considerable carbon costs (Figure 6). In particular, the FW + HM combination at an input rate of 50 g pot$^{-1}$ and the single FW at 100 g pot$^{-1}$ both display substantial carbon costs. These costs are notably higher when compared to the FW + HM combination at a lower input rate of 25 g pot$^{-1}$. However, it's important to note that no significant interaction was observed in any of the treatments across all input rates.

All treatments applied to Komatsuna plants had a significant impact on the nitrogen content percentage. The highest nitrogen content percentage was observed when using a fertilizer rate of 100 g pot$^{-1}$. Furthermore, when considering treatments based on the type of fertilizer, the highest yield was achieved with the single FW compost (refer to Table 3). Similarly, in Komatsuna media treated with a fertilizer dosage of 100 g pot$^{-1}$, the highest nitrogen percentage was achieved, with the treatment based on the type of fertilizer also yielding the highest output, specifically with the single FW compost (refer to Table 3).

The data illustrated in Figure 7 reveal that the maximum uptake of N is achieved when fertilizer is applied at a rate of 100 g pot$^{-1}$. However, this uptake varies depending on the type of plant. The combination of FW + HM compost, when applied at the same rate, proved to be the most efficient fertilizer in terms of N uptake. Figure 8, on the other hand, demonstrates that the efficiency of NUE, when evaluated based on the fresh weight

of the plant, showed optimal results with Komatsuna plants. These plants were treated with a combination of FW and HM compost, but at a lower rate of 25 g pot$^{-1}$. It's crucial to understand that the effectiveness of methane-fermented fertilizer is not solely due to its inherent properties. The addition of manure significantly influences its efficacy, which subsequently impacts crop yields. Moreover, the quality of methane-fermented fertilizer can be affected by the type of feed given to livestock, a point further expounded by [4].

This study found that the most effective dose for mixing fertilizer was 100 g pot$^{-1}$ in all treatments, except for single FW compost. This finding is consistent with the research conducted by [4], which suggests that growth inhibition can occur due to the high N content in both fertilizers. The nutritional composition of compost offers highly favorable nutrition for plants. Elevated levels of nutrients, particularly N, exert a significant influence on plant growth, posing the risk of stunting and impacting nutrient accumulation in the soil. This aligns with findings reported [31], which emphasize the critical importance of the release rate or availability of nitrogen. The evaluation of in situ nitrogen mineralization is proposed as a means to enhance NUE [32]. reported an increase in N input through high N-fixation rest, which resulted in a higher availability of N for the crop and improved NUE. Nitrogen plays a crucial role in plant growth as a structural element. In roots, these elements exist as proteins and enzymes that facilitate the absorption of water and nutrients for plant needs. It's important to note that an excessive application of N-containing fertilizers does not necessarily lead to increased plant growth. The N content in plants varies across species, but typical concentrations range from 1.5% to 6% of the dry weight of many plants. Optimal values are typically found between 2.5% and 3.5% in leaf tissue [33,34].

The significant impact of increased N uptake and heightened carbon content in Komatsuna plants on the growth process is evident in their robust production. Overall, treatments with crops exhibiting elevated N uptake and carbon content, especially at the input level of 100 g pot$^{-1}$, effectively demonstrated a substantial influence on crop yields. Enhanced plant quality serves as an indicator of favorable plant nutrition. However, it is important to note that this research did not specifically assess the overall nutritional value and health of the plants, including potential negative aspects.

Figure 9, which illustrates the cost performance of fertilizer, underscores the impact of biochar on the cost-effectiveness of fertilizer. It's been observed that the unit cost of commercially available biochar is higher than that of homemade biochar. This difference significantly affects the FCP when using commercial biochar (Figure 9B) as compared to homemade biochar (Figure 9A). The FCP analysis in this research adopts a long-term economic perspective by examining fluctuations in fertilizer costs within a specific area. This includes assessing the quantity of fertilizer input applied to the land, the prevailing unit price of fertilizer in the market, and the corresponding harvest outcomes achieved in each treatment.

Figure 10 presents a comparison of the BC of B/C ratio and economic costs for homemade (Figure 10A) and commercial (Figure 10B) products. In the analysis of economic implications associated with the integration of biochar, a simulation of agricultural conditions is utilized. This comprehensive analysis considers the utilization of both commercial and homemade biochar, taking into account various factors such as costs, revenues, and cash flows involved in farming practices. The calculation of cost factors involves aggregating costs per planting season per hectare unit. This calculation encompasses material costs, daily labor expenses at an hourly rate, and transportation expenses for materials per planting season. There is a significant difference between these two, which is influenced by the profit margins and total costs associated with commercial biochar products. The total cost of commercial biochar itself is influenced by various factors, including material and electrical costs, which contribute to higher variable costs compared to homemade biochar.

## 5. Conclusions

The germination of Komatsuna showed a significant improvement when the FW and BC compost combination was used, outperforming other compost mixtures. This strongly

suggests that the use of methane-fermented compost along with BC can greatly enhance the growth of Komatsuna seeds. Moreover, the combined use of biochar and compost shows great potential for increasing the yield of Komatsuna crops while also increasing the overall nitrogen content in both plants and soil. It's worth noting that the addition of biochar had a considerable impact on the interaction between input rate, compost, and biochar in terms of N uptake, soil carbon, and Komatsuna yield. However, such an effect was only observed in the interaction between biochar and compost in terms of NUE. It's also important to highlight that homemade BC shows a decreasing trend in FCP as input rates increase across all fertilizer combinations. In contrast, commercially priced BC shows an increasing trend in FCP as input rates decrease across all fertilizer mixtures. These findings indicate that the biochar enhances the activity of methane fermentation residue compost thereby promoting plant growth through environmentally sustainable waste processing. The proposed mixture can potentially be utilized for cultivating various types of plants. However, it's crucial to note that the nutritional content of the compost produced in each location may vary, making general comparisons difficult. Nonetheless, careful application of the inputs during planting is essential. This research holds particular relevance for crops emphasizing leavy vegetable plants. Limitations primarily revolve around the diligence applied to compost application, encompassing both field and greenhouse maintenance. While this study is significant for sustainable agricultural practices, it's important to acknowledge its limitations. One notable constraint is the lack of qualitative analysis to evaluate the potential environmental benefits based on life cycle thinking. Addressing this limitation is imperative, and future research endeavors will aim to rectify this gap.

**Author Contributions:** Conceptualization, M.K., N.S., S.W., Y.S. and T.I.; methodology, M.K.; validation, M.K.; formal analysis, N.S.; writing—original draft preparation, N.S.; writing—review and editing, R.K.D.; visualization, N.S. and R.K.D. All authors have read and agreed to the published version of the manuscript.

**Funding:** This research received was funded by grants from Hitachi Cement Co., Ltd.

**Institutional Review Board Statement:** Not applicable.

**Informed Consent Statement:** Not applicable.

**Data Availability Statement:** Data is contained within the article.

**Acknowledgments:** This work was supported by the Hitachi Cement Co., Ltd and Life Earth Agricultural Future Co., Ltd.

**Conflicts of Interest:** The authors declare no conflict of interest. The funding sponsors had no role in the design of the study; in the collection, analyses, or interpretation of data; in the writing of the manuscript, and in the decision to publish the results.

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
