# Peer review of "Enhancing Sustainable Waste Management Using Biochar: Mitigating the Inhibitory of Food Waste Compost from Methane Fermentation Residue on Komatsuna (Brassica rapa) Yield"

_sustainability, doi:10.3390/su16062570_

Round 1
Reviewer 1 Report
Comments and Suggestions for Authors
The paper presents a valuable study on the use of methane fermentation, utilizing food waste (FW), as a sustainable strategy for environmental conservation. The study aims to assess the influence of various compost mixtures on the germination, growth, and yield of Komatsuna (Brassica rapa), employing a completely randomized design with biochar (BC), clay (CL), weeds (WD), and a control group. The experimental setup, involving a complete factorial design with different compost types, biochar presence, and input rates, provides a comprehensive evaluation of the effects on plant growth.
However, the clarity regarding the novelty of this study is somewhat obscured in the abstract and introduction. The quantitative results presented do not sufficiently support the study's conclusions.
Here are suggestions for enhancement :
-
1. The selection of materials from a limited geographical area raises questions about the generalizability of the findings. Considering the variability in food waste composition due to regional dietary differences, expanding the material selection to encompass a wider range of locations could enhance the study's applicability and reliability.
-
2. While the results section is rich with diagrams, a concise explanation accompanying each would facilitate a quicker and deeper understanding by readers, linking the visual data directly to the study's findings. This approach would significantly improve the readability and accessibility of the research.
-
3. The conclusion's discussion on the measurement of plant radicle length requires further elaboration to elucidate its relevance and the conclusions derived from these measurements, thereby providing a clearer understanding of the study's outcomes.
-
4. The methodology section should emphasize ensuring consistent germination and growth conditions across all experimental groups to minimize external influences. Expanding the control group sample size and increasing the number of experimental repetitions would bolster the study's reliability.
-
5. The economic implications of composting, particularly the cost/benefit analysis concerning biochar addition, warrant a more detailed examination. While biochar's benefits for plant growth are acknowledged, the analysis should also consider the economic feasibility, especially given the potential cost implications of biochar use.
- 6. At least, qualitative analysis should be carried out for eavluating the potential envrionemtnal impacts and benefits for this process based on life-cycle thinking. This analysis should cover all stages from production and usage to disposal, thereby identifying potential areas for environmental improvement and sustainability enhancement. It would not only enrich the study's findings but also align with the growing emphasis on sustainable practices in waste management.
Author Response
Reviewer #1
The paper presents a valuable study on the use of methane fermentation, utilizing food waste (FW), as a sustainable strategy for environmental conservation. The study aims to assess the influence of various compost mixtures on the germination, growth, and yield of Komatsuna (Brassica rapa), employing a completely randomized design with biochar (BC), clay (CL), weeds (WD), and a control group. The experimental setup, involving a complete factorial design with different compost types, biochar presence, and input rates, provides a comprehensive evaluation of the effects on plant growth.
However, the clarity regarding the novelty of this study is somewhat obscured in the abstract and introduction. The quantitative results presented do not sufficiently support the study's conclusions.
Here are suggestions for enhancement :
- The selection of materials from a limited geographical area raises questions about the generalizability of the findings. Considering the variability in food waste composition due to regional dietary differences, expanding the material selection to encompass a wider range of locations could enhance the study's applicability and reliability.
(Comments)
Regarding the variability in food waste composition, we have shown the comparison of nutritional quality of the compost based on NPK levels and applied it based on the common application rate. (in line 139-149)
(Revision)
- While the results section is rich with diagrams, a concise explanation accompanying each would facilitate a quicker and deeper understanding by readers, linking the visual data directly to the study's findings. This approach would significantly improve the readability and accessibility of the research.
(Comments)
Thank you for your suggestions. In this manuscript, I divided the discussion best on the germination and pot experiment. They have a specific variable to discuss, and I have explain all of the relationships among variable accordingly.
(Revision)
- The conclusion's discussion on the measurement of plant radicle length requires further elaboration to elucidate its relevance and the conclusions derived from these measurements, thereby providing a clearer understanding of the study's outcomes.
(Comments)
The radicle length during germination serves as evidence for identifying potential growth-inhibiting factors. Therefore, measuring radicle length is imperative in assessing the effectiveness of the germination process.
(Revision)
- The methodology section should emphasize ensuring consistent germination and growth conditions across all experimental groups to minimize external influences. Expanding the control group sample size and increasing the number of experimental repetitions would bolster the study's reliability.
(Comments)
The germination was conducted in the control condition particularly we incubated the seed at a temperature of 24 °C. The pot experiment also conducted in the greenhouse that controlled to minimize the external influences. We have emphasized this methodology in the manuscript following this sentence,
… In order to minimize the external influences, the compost germination test was conducted in a laboratory setting, utilizing a completely randomized experimental design with three replications. …
… A greenhouse pot experiment was carried out over a period of one month also under controlled condition. …
(Revision)
- The economic implications of composting, particularly the cost/benefit analysis concerning biochar addition, warrant a more detailed examination. While biochar's benefits for plant growth are acknowledged, the analysis should also consider the economic feasibility, especially given the potential cost implications of biochar use.
(Comments)
The economic feasibility was evaluated using a benefit and cost ratio (B/C) ratio. When the B/C ratio was more than 1, it means that the management is feasible. It can be found in the homemade biochar.
In the analysis of economic implications associated with the integration of biochar, a simulation of agricultural conditions is employed. The comprehensive analysis encompasses the utilization of both commercial and homemade biochar, considering various factors such as costs, revenues, and cash flows involved in farming practices. The calculation of cost factors, involves aggregating costs per planting season per hectare unit. This calculation encompasses material costs, daily labor expenses per hourly costs, and transportation expenses for materials per planting season.
(Revision)
- At least, qualitative analysis should be carried out for evaluating the potential environmental impacts and benefits for this process based on life-cycle thinking. This analysis should cover all stages from production and usage to disposal, thereby identifying potential areas for environmental improvement and sustainability enhancement. It would not only enrich the study’s findings but also align with the growing emphasis on sustainable practices in waste management.
(Comments)
I recognize the imperative for qualitative analysis; however, it has not been conducted in this research. Perhaps in future research we will incorporate qualitative analysis, specifically employing a life cycle thinking framework, as a method for evaluating the potential and impact of environmental benefits based on social perspective.
(Revision)

Reviewer 2 Report
Comments and Suggestions for Authors
Please find it attached.

Author Response
Reviewer#2
¥
Review Report Form:
Enhancing Sustainable Waste Management Using Biochar: Mitigating the Inhibitory of
Food Waste Compost from Methane 3 Fermentation Residue on Komatsuna (Brassica rapa)
Yield.
Table 1, 2, 3. Incorrect format because the lines in borders are thick.
(Comments)
The manuscript has been revised based on the comments.
(Revision)
Figure 1. The text in color white can not be looks fine, the suggestions is that the text can be
bellow in color black.
(Comments)
The manuscript has been revised based on the comments.
(Revision)
The equations do not have the correct format.
Figure 2. The R2 can be closer than 1.0, suggestion the authors can do another calibration
curve.
(Comments)
Figure 2, NH4+correlation with germination index have a R2 0.9001, and it have a significant correlation.
(Revision)
Figures 2,3. The color orange can be change for grey and the letters inside the bars can be
showed better.
(Comments)
The manuscript has been revised based on the comments.
(Revision)
Figure 4, 5, 6, 7, 8. The letters that indicates significant differences can be smaller so they
can be look better
(Comments)
The manuscript has been revised based on the comments.
(Revision)
Additional comments of the paper: Chemical Characterization of Red Wine Polymers and
Their In-2teraction Affinity with Odorants
- What is the main question addressed by the research?
The authors describe analyze then compounds that gave aroma to red wine
- What parts do you consider original or relevant for the field? What specific gap in the field
does the paper address?
-The different degradation methods (sulfuric acid and hydrochloric Acid, alkaline method)
used by authors to obtained fractions of the aroma compounds. They do a comparation
among these methods and the results of the fraction obtained from them.
- What does it add to the subject area compared with other published materials?
-The authors not only used different methods to obtained the fraction of wine also they do
an identification by chromatography technics and also they complete the paper with the
sensory analysis.
- What specific improvements should the authors consider regarding the methodology?
What further controls should be considered?
-Maybe the authors can improve the experiment doing an compounds extraction using a
green technology (thermal process, ultrasound, microwave) and do the comparation among
green technology and chemical methodology.
- Please describe how the conclusions are or are not consistent with the evidence and
arguments presented. Please also indicate if all main questions posed were addressed and
2
by which specific experiments.
-The conclusions section is almost complete because the authors do not express a conclusion
of the sensory analysis, that part is missing.
- Are the references appropriate?
-The references are related to the paper and the information is actual except for the
references 14 (2000), 16 (2000), 26 (1979), 29 (1972), 47 (1957)
- Please include any additional comments on the tables and figures and quality of the data.
-Figure 1 is very explicit about what the methods that the authors used to obtained the
results about aroma fractions of the red wine; however I suggest that the methods can
be write in capital letter as: Alkaline, Thiolysis. Figure 2 the chemical formula of the
compound on the top of the qHNMR spectra is sma
(Comments)
The comments were not relevant to this manuscript.
(Revision)

Reviewer 3 Report
Comments and Suggestions for Authors
The article Enhancing Sustainable Waste Management Using Biochar: Mitigating the Inhibitory of Food Waste Compost from Methane Fermentation Residue on Komatsuna (Brassica rapa) Yield proposes a method of managing food waste as a raw material in the methane fermentation process, using admixtures of natural origin. The aim of the study was to assess the effect of prepared compost mixtures on the increase in seed germination and the yield of Brassica rapa..
I fully agree with the formal approach to the research problem and the way of solving it, including the adopted methodology of the experiment and the method of conducting it.
The issue raised by the Authors is an interesting example of the use of organic carbon in solving practical problems in the field of plant cultivation and environmental protection. For this reason, I would like to recommend the submitted manuscript for publication, with reference to the following questions and comments:
1. Is it possible to use the proposed mixtures in the cultivation of other plant species, especially those found in other climatic zones?
2. What does food waste consist of (line 88)?
3. Please standardize the units of mass (megagrams and tons) in table 2.
4. Equations 1 and 2 should be presented in a professional manner. What units are the dependent variables in formulas 3 and 4? Can the "=" sign in a scientific text have non-mathematical uses (line 197)?
5. Please describe the axes (units) of the plots in Figures 2, 5, 6 and 8. Are all the graph axes in Figures 2 and 10 rational and legible?
6. What do the lowercase Latin letters in the last two columns of Table 3 (top), also present in the graphs (Figures 3–8) mean?
7. Please correct the chemical notation (lines 242, 243, 249 and others, table 2).
8. Please remove unnecessary repetition of literature sources (lines 58, 59, 65 and 67).
9. The last paragraph (conclusions) requires strengthening. In what direction can the research go?
In my opinion, the article meets the high standards of the Journal. For this reason, I recommend the publication of the reviewed article in the Sustainability after addressing the above questions and comments.

Author Response
Reviewers #3
The article Enhancing Sustainable Waste Management Using Biochar: Mitigating the Inhibitory of Food Waste Compost from Methane Fermentation Residue on Komatsuna (Brassica rapa) Yield proposes a method of managing food waste as a raw material in the methane fermentation process, using admixtures of natural origin. The aim of the study was to assess the effect of prepared compost mixtures on the increase in seed germination and the yield of Brassica rapa..
I fully agree with the formal approach to the research problem and the way of solving it, including the adopted methodology of the experiment and the method of conducting it.
The issue raised by the Authors is an interesting example of the use of organic carbon in solving practical problems in the field of plant cultivation and environmental protection. For this reason, I would like to recommend the submitted manuscript for publication, with reference to the following questions and comments:
- Is it possible to use the proposed mixtures in the cultivation of other plant species, especially those found in other climatic zones?
(Comments)
Yes, it is possible to utilize the suggested mixture for cultivating other plant species especially for the leafy vegetable that has similar characteristics. The cultivation method is applicable even in other climatic zone since we can control it with the green house. However, it is imperative to note that the nutritional content of the compost generated in each location will be different, so it is important to adjust the application rate based on the NPK level of compost.
(Revision)
- What does food waste consist of (line 88)?
(Comments)
The manuscript has been revised based on the comments.
Food waste comprises household waste, discarded items from restaurants, and excess products from surrounding markets near waste processing facilities. Prior to its utilization in a biogas plant, the waste undergoes an initial filtration process to eliminate extraneous materials, including plastic bottles, metal, and glass. Subsequently, the filtered waste is shredded into smaller particles, and the organic components are introduced into a fermenter tank containing methanogenic bacteria.
(Revision)
- Please standardize the units of mass (megagrams and tons) in table 2.
(Comments)
The unit of anion and cation in table 2 is in mg kg-1 (milligram of anion, cation or per kilogram of soil) not megagrams or tons.
(Revision)
- Equations 1 and 2 should be presented in a professional manner. What units are the dependent variables in formulas 3 and 4? Can the "=" sign in a scientific text have non-mathematical uses (line 197)?
(Comments)
The manuscript has been revised based on the comments.
(Revision)
- Please describe the axes (units) of the plots in Figures 2, 5, 6 and 8. Are all the graph axes in Figures 2 and 10 rational and legible?
(Comments)
The manuscript has been revised based on the comments.
GI and NUE don’t have units, figures 2 and 10 have rational and legible.
(Revision)
- What do the lowercase Latin letters in the last two columns of Table 3 (top), also present in the graphs (Figures 3–8) mean?
(Comments)
The manuscript has been revised based on the comments.
Value = mean ± standard error. The presence of different lowercase letters indicates significant differences between the compost treatments, input rates, and biochar treatments different at based Tukey–Kramer test.
(Revision)
- Please correct the chemical notation (lines 242, 243, 249 and others, table 2).
(Comments)
The manuscript has been revised based on the comments.
(Revision)
- Please remove unnecessary repetition of literature sources (lines 58, 59, 65 and 67).
(Comments)
The manuscript has been revised based on the comments.
(Revision)
- The last paragraph (conclusions) requires strengthening. In what direction can the research go?
(Comments)
These findings indicate that the incorporation of biochar improves the efficacy of methane fermentation residue compost, subsequently development plant growth through the utilization of environmentally sustainable waste processing.
(Revision)
In my opinion, the article meets the high standards of the Journal. For this reason, I recommend the publication of the reviewed article in the Sustainability after addressing the above questions and comments.

Reviewer 4 Report
Comments and Suggestions for Authors
First of all, I would like to thank you very much for choosing our journal for your article. It is a very successful and meticulously prepared article. If you answer the questions I have asked, I would like to read the article again.
- Could you provide more details on the criteria used for selecting the locations in Tsuchiura City from where the food waste was collected? How representative are these locations of the overall waste profile of the city?
- Can you elaborate on the specific challenges faced by the compost in terms of imbalanced nutrient content, unpleasant odor, and growth inhibition? How does your research address these challenges?
- Are there plans for further research to optimize the compost mix or to explore its application on other crops besides Komatsuna? What future directions do you suggest for research in this field?
- Regarding the assessment of compost quality (Section 2.2), can you provide more insight into the selection of the methods used, such as ion chromatography for quality evaluation? How do these methods compare with other possible techniques?
- How does your research contribute to the sustainable agriculture practices in Ibaraki Prefecture? Are there any specific policy implications or recommendations for local farmers or authorities?
- How applicable are your findings to other regions or countries with similar agricultural and waste management profiles? Are there any limitations to the generalizability of your results?
- Could you provide more details on how the “Input N” was calculated for each treatment? What specific forms of nitrogen were applied, and were they consistent across all treatments?
- For the FCP analysis, how did you account for variations in fertilizer costs over time or across different regions? Does this analysis consider long-term economic sustainability?
- In your results on Komatsuna seed germination, you observed different germination rates and radicle lengths with various compost mixtures. How do these findings contribute to current knowledge in the field, and what are their practical implications for agricultural practices?
- Your study indicates a negative correlation between ammonia content and germination index. Could you discuss the potential reasons behind this observation and its implications for compost formulation?
- Regarding the carbon and nitrogen uptake in Komatsuna, how does the increased uptake affect the overall health and nutritional value of the plants? Are there any trade-offs to consider?
- How applicable are your findings to other crops or different agricultural contexts? Are there any limitations or specific conditions under which your results may vary?
- In addition to food, a brief mention of sustainable energy sources will increase the depth of your article. Please include these two studies in the introduction.
Comparison of the Techno-Economic and Environmental Assessment of Hydrodynamic Cavitation and Mechanical Stirring Reactors for the Production of Sustainable Hevea brasiliensis Ethyl Ester
https://doi.org/10.3390/su152316287
Covid-19 and the politics of sustainable energy transitions
https://doi.org/10.1016/j.erss.2020.101685
- Could you explain the methodology used for the comprehensive cost-benefit analysis of compost utilization? How did you account for different cost factors such as material, labor, and transportation in your analysis?
- In your economic analysis, how did you evaluate the cost-effectiveness of combining different types of compost, such as FW with BC, HM with BC, and FW with HM with BC? What specific factors were considered in these evaluations?
- In Figure 10, you present an economic cost analysis of biochar systems. Can you provide more details on how you calculated the profit margins and benefit-to-cost ratios for homemade vs. commercial biochar? What assumptions were made in this analysis?
- You discuss the impact of nitrogen content on plant growth. Could you elaborate on how different nitrogen levels in the compost affect the nutritional value and overall health of the Komatsuna plants?
Author Response
Reviewers #4
First of all, I would like to thank you very much for choosing our journal for your article. It is a very successful and meticulously prepared article. If you answer the questions I have asked, I would like to read the article again.
- Could you provide more details on the criteria used for selecting the locations in Tsuchiura City from where the food waste was collected? How representative are these locations of the overall waste profile of the city?
(Comments)
These areas serve as facilities for processing methane fermentation bioplants, recycling food-related waste. The biogas produced acts as additional fuel for eco-plants, like power facilities particularly in the Ibaraki prefecture region.
(Revision)
- Can you elaborate on the specific challenges faced by the compost in terms of imbalanced nutrient content, unpleasant odor, and growth inhibition? How does your research address these challenges?
(Comments)
The challenge of this research is to overcome the methane fermentation residue compost that have imbalanced nutritional composition and hinders plant growth from germination to harvest upon application. Additionally, the potent odor associated with the compost diminishes farmers' enthusiasm for its utilization. This study showers the combination of methane fermentation foot waste residue compost with biochar could enhance the germination and reduce the odor.
(Revision)
- Are there plans for further research to optimize the compost mix or to explore its application on other crops besides Komatsuna? What future directions do you suggest for research in this field?
(Comments)
Yes, in addition to our ongoing research on cabbage and sorghum, we are engaged in investigating environmental pollution through the observation of greenhouse gas emissions. In future studies, we aim to adopt an innovative approach to mixing by utilizing specialized mixing machines rather than traditional methods. Our aspiration is that the enhanced sustainability among the interconnected facets of this research will play a crucial role in optimizing waste utilization and addressing global waste processing challenges.
(Revision)
- Regarding the assessment of compost quality (Section 2.2), can you provide more insight into the selection of the methods used, such as ion chromatography for quality evaluation? How do these methods compare with other possible techniques?
(Comments)
Ion chromatography, also known as ion-exchange chromatography, separates ions and polar molecules based on their affinity to the ion exchanger. An alternative involves using an ion chromatograph with a PU-2080i plus HPLC pump, although it represents an older version compared to the Eco IC
(Revision)
- How does your research contribute to the sustainable agriculture practices in Ibaraki Prefecture? Are there any specific policy implications or recommendations for local farmers or authorities?
(Comments)
This research supports sustainable agriculture in Ibaraki Prefecture, aligning with the Green Food System Law for Promoting Prefectural Environmental Burden Reducing Business Activities,' released in March 2023.
(Revision)
- How applicable are your findings to other regions or countries with similar agricultural and waste management profiles? Are there any limitations to the generalizability of your results?
(Comments)
This research is very relevant to various regions in any part of the country that experience waste processing problems, it's just that organic waste processing will greatly influence the results, especially on the nutritional content of the compost produced, so in each region this is a limitation of this research.
(Revision)
- Could you provide more details on how the “Input N” was calculated for each treatment? What specific forms of nitrogen were applied, and were they consistent across all treatments?
(Comments)
The calculated nitrogen input in each treatment is contingent upon the compost input levels employed in this study: 25 g pot-1, 50 g pot-1, and 100 g pot-1. This input is determined by multiplying the respective nitrogen content percentages for each compost type. The nitrogen is present in the form specific to the nutrients found in each compost. The consistent application of nitrogen across all treatments is based on the compost type employed in each respective treatment.
(Revision)
- For the FCP analysis, how did you account for variations in fertilizer costs over time or across different regions? Does this analysis consider long-term economic sustainability?
(Comments)
The FCP analysis in this research incorporates a long-term economic perspective by assessing fluctuations in fertilizer costs within a given area. This involves evaluating the quantity of fertilizer input applied to the land, the prevailing unit price of fertilizer in the market, and the corresponding harvest outcomes achieved in each treatment.
(Revision)
- In your results on Komatsuna seed germination, you observed different germination rates and radicle lengths with various compost mixtures. How do these findings contribute to current knowledge in the field, and what are their practical implications for agricultural practices?
(Comments)
The optimal outcomes in the germination test within this study demonstrated that the amalgamation of compost generated through methane fermentation with biochar effectively mitigated the growth inhibitors present in the methane fermentation-produced compost. This suggests that such a combination can harmonize the nutrient profile of the compost, offering farmers a more accessible choice of relatively affordable alternatives.
(Revision)
- Your study indicates a negative correlation between ammonia content and germination index. Could you discuss the potential reasons behind this observation and its implications for compost formulation?
(Comments)
The germination outcomes of methane fermentation-derived compost indicate the presence of growth inhibitors associated with elevated ammonia content. Specifically, 1 mM ammonium negatively affects primary root growth, elemental expansion, cell production, and maximum elemental expansion rate, inhibits root gravitropism and concomitantly down-regulates the expression of two pivotal auxin transporters, AUX1 and PIN2.
(Revision)
- Regarding the carbon and nitrogen uptake in Komatsuna, how does the increased uptake affect the overall health and nutritional value of the plants? Are there any trade-offs to consider?
(Comments)
Overall treatment of crop yields that have high N uptake content and carbon content effectively have a high influence on the input level of 100 g pot-1. Increasing N uptake and carbon content in komatsuna plants greatly influences the growth process so that it the high production of plants, the better the quality of the plants produced will indicate that the plant nutrition is also good, but this research did not specifically observe the nutritional value and health of the plants as a whole or has excessive negative content for plants.
(Revision)
- How applicable are your findings to other crops or different agricultural contexts? Are there any limitations or specific conditions under which your results may vary?
(Comments)
"This research holds high relevance for any crop centered on green vegetable plants. The limitations are contingent upon the attention dedicated to the application of compost to plants, encompassing both field and greenhouse maintenance."
(Revision)
- In addition to food, a brief mention of sustainable energy sources will increase the depth of your article. Please include these two studies in the introduction.
Comparison of the Techno-Economic and Environmental Assessment of Hydrodynamic Cavitation and Mechanical Stirring Reactors for the Production of Sustainable Hevea brasiliensis Ethyl Ester
https://doi.org/10.3390/su152316287
Covid-19 and the politics of sustainable energy transitions
https://doi.org/10.1016/j.erss.2020.101685
(Comments)
I have added the following sentence in the introduction section.
…[5] report, the primary objectives include enhancing agricultural practices, promoting clean energy, and mitigating carbon emissions. According [6], as outlined in the Paris Agreement, nationally determined climate goals initiated sustainability transitions on the agendas of numerous local, national, and global governing bodies. Concurrently, the cost of renewable energy experienced a rapid decline, rendering it a progressively politically and economically viable option. Additionally, there was a notable increase in public support for the imperative need for urgent action to address climate change. This constitutes a critical aspect of research that aligns with the prioritization of environmental issues in accordance in this research objectives…
(Revision)
- Could you explain the methodology used for the comprehensive cost-benefit analysis of compost utilization? How did you account for different cost factors such as material, labor, and transportation in your analysis?
(Comments)
The cost factors involved in the calculation, including materials, labor, and transportation, are computed by aggregating the costs per planting season per hectare unit. This encompasses the material costs necessary for one planting season, the labor expenses incurred per day, and the hourly work costs, along with the transportation expenses for materials per planting season.
(Revision)
- In your economic analysis, how did you evaluate the cost-effectiveness of combining different types of compost, such as FW with BC, HM with BC, and FW with HM with BC? What specific factors were considered in these evaluations?
(Comments)
In the assessment of economic analysis, we compared harvest yields across treatments and multiplied the unit price for each type of fertilizer. The analysis includes factors such as the unit price of fertilizer and the yield of each treatment relative to the control yield. A crucial consideration is quantifying the fertilizer input provided to the plant in relation to the achieved harvest. This evaluation enables us to gauge the effectiveness of fertilizer utilization in terms of performance and cost.
(Revision)
- In Figure 10, you present an economic cost analysis of biochar systems. Can you provide more details on how you calculated the profit margins and benefit-to-cost ratios for homemade vs. commercial biochar? What assumptions were made in this analysis?
(Comments)
In the examination of the economic implications associated with the incorporation of biochar, we employ a simulation of agricultural conditions. This simulation replicates the planting procedures conducted within a controlled greenhouse on a 1-hectare agricultural land area. The analysis encompasses the utilization of both commercial and homemade biochar, taking into account diverse factors such as costs, revenues, and cash flows involved in the farming practices.
(Revision)
- You discuss the impact of nitrogen content on plant growth. Could you elaborate on how different nitrogen levels in the compost affect the nutritional value and overall health of the Komatsuna plants?
(Comments)
The manuscript has been revised based on the comments.
The considerable influence of heightened nitrogen (N) uptake and increased carbon content in komatsuna plants on the growth process is evident in the high production of plants. Overall, treatments featuring crops with elevated nitrogen uptake and carbon content, particularly at the input level of 100 g pot-1, effectively demonstrated a substantial impact on crop yields. Improved plant quality serves as an indicator of favorable plant nutrition. Nevertheless, it is noteworthy that this research did not specifically assess the overall nutritional value and health of the plants, encompassing potential negative aspects.
(Revision)

Round 2
Reviewer 1 Report
Comments and Suggestions for Authors
The revised paper can be published.
Author Response
There is nothing to revise based on reviewer 1.
Thank you
Reviewer 4 Report
Comments and Suggestions for Authors
Accept in present form.
Author Response
Response for Reviewers
Reviewer #4
(Comments)
Is the content succinctly described and contextualized with respect to previous and present theoretical background and empirical research (if applicable) on the topic?
(Revision)
… These challenges are closely related to the increase in organic waste…
…To overcome these issues, by encouraging sustainable food production through a combination of the use of organic resources as an important part…
(Comments)
Are the arguments and discussion of findings coherent, balanced and compelling?
(Revision)
I corrected the part of the paragraph that was placed incorrectly in the discussion of carbon, it will be the arguments and discussion of findings balanced and compelling.
…When it comes to carbon cost analysis, the results mirror the levels of carbon in the soil. Notably….
(Comments)
For empirical research, are the results clearly presented?
(Revision)
I corrected the graph for part 10b, there is a slight improvement on the y-axis of the number categories because previously it was not the same as part 10a.
(Comments)
Are the conclusions thoroughly supported by the results presented in the article or referenced in secondary literature?
(Revision)
Yes, sure. These findings indicate that the biochar enhances the activity of methane fermentation residue compost thereby promoting plant growth through environmentally sustainable waste processing.
